# Natural variation of macrophage activation as disease-relevant phenotype predictive of inflammation and cancer survival

Konrad Buscher[1,*], Erik Ehinger[1,*], Pritha Gupta[2,*], Akula Bala Pramod[1], Dennis Wolf[1], George Tweet[1], Calvin Pan[2], Charles D. Mills[3,‡], Aldons J. Lusis[2] & Klaus Ley[1]

Although mouse models exist for many immune-based diseases, the clinical translation remains challenging. Most basic and translational studies utilize only a single inbred mouse strain. However, basal and diseased immune states in humans show vast inter-individual variability. Here, focusing on macrophage responses to lipopolysaccharide (LPS), we use the hybrid mouse diversity panel (HMDP) of 83 inbred strains as a surrogate for human natural immune variation. Since conventional bioinformatics fail to analyse a population spectrum, we highlight how gene signatures for LPS responsiveness can be derived based on an Interleukin-12β and arginase expression ratio. Compared to published signatures, these gene markers are more robust to identify susceptibility or resilience to several macrophage-related disorders in humans, including survival prediction across many tumours. This study highlights natural activation diversity as a disease-relevant dimension in macrophage biology, and suggests the HMDP as a viable tool to increase translatability of mouse data to clinical settings.

[1] Division of Inflammation Biology, La Jolla Institute for Allergy and Immunology, La Jolla, California 92037, USA. [2] Departments of Medicine, Human Genetics, and Microbiology, Immunology, and Molecular Genetics, University of California, Los Angeles, California 90095, USA. [3] BioMedical Consultants, Marine on St. Croix, Minnesota 55047, USA. * These authors contributed equally to this work. Correspondence and requests for materials should be addressed to K.L. (email: klaus@lji.org).
‡Deceased.

Clinical and pharmaceutical researchers are concerned with the lack of relevance and low reproducibility of findings obtained in standard mouse models[1–3]. The translation of mouse data from bench to bedside is challenging[4–8] and clinical trial success rates continue to be low[1,9]. Among the possible reasons, the large inter-individual variation of immune system variables in human populations is cited[10,11]. This natural variation has been shown to broadly impact on pathophysiology, for example, disease resilience, tolerance[12–14] and vaccination responses[10,15]. System level analyses of identical twins find both environmental and genetic components of variability[16]. Recently, inbred mouse strains were placed in a more natural, 'dirty' environment leading to greater variation of immune cell populations[17]. Here, we focus on how introducing genetic diversity into mouse research reveals a much broader spectrum of innate immune responses.

Macrophages are widely distributed throughout the body and are hence one of the first cells to react to a perturbation of homoeostasis. Their functional programs are highly context-dependent and mostly influenced by pathogens, cellular origin (monocyte-derived or embryonic) and tissue cues (including cytokines)[18,19]. Among a continuum of stimulus-dependent polarization states[18,20–23], most with unknown function, M1 reflects a pro-inflammatory phenotype with pathogen killing abilities and can be induced by lipopolysaccharide (LPS), and M2 reflects the default state of tissue macrophages that promotes homeostasis and wound healing. These macrophages metabolize arginine to citrulline and nitric oxide through inducible nitric oxide synthase (iNOS, Nos2). The production of the macrophage cytokine Interleukin-12 (IL-12) is a hallmark of M1 and promotes a Th1 response[24,25]. M2 macrophages can metabolize arginine to ornithine, a precursor of polyamine and hydroxyproline, and urea through Arginase (Arg1), which promotes wound healing, angiogenesis and tissue homoeostasis[24,25]. iNOS is a crucial regulator in mouse M1 macrophages, but its relevance in human macrophages is still debated[26,27]. Importantly, genetic variation has been shown to affect macrophage activation by hierarchical functions of lineage-determining transcription factors[28]. While significant differences in Th1/2 polarization propensity in healthy individuals has been shown[29], little is known about natural diversity of macrophage polarization in health and disease.

The hybrid mouse diversity panel (HMDP) is a panel of about 100 inbred mouse strains developed for performing association studies with adequate statistical power and resolution for mapping of complex traits[30,31]. It has been successfully used for investigating gene-environment interaction in activated macrophages[32], insulin sensitivity[33], and susceptibility to atherosclerosis[34]. Here, we employ the HMDP as a surrogate model for human immune diversity and investigate how meaningful gene signatures of macrophage activation can be extracted from a heterogeneous population. We demonstrate that these signatures are highly robust in predicting disease susceptibility and outcomes in humans, suggesting that immune diversity is a critical parameter in translational medicine.

## Results

**Natural variation of LPS activation in macrophages**. LPS is found in gram-negative bacterial membranes and elicits strong inflammatory immune responses, mainly via Toll-like receptor (TLR) 4 that induces activation of the NF-κB pathway[35]. To estimate the inter-individual variation of macrophage LPS responses in a diverse population we compared the transcriptional activation patterns in humans and mice. In human alveolar macrophages exposed to LPS *in vivo* we observed large intrinsic variation in inter-individual genetic responses, for example, NF-κB and TLR pathway activities (Fig. 1a,b). In thioglycollate-elicited peritoneal macrophages of HMDP strains exposed to LPS, we found a similarly large range of pathway activations (Fig. 1c,d). Notably, LPS incubation was short (4 h), and reflects an early state of macrophage activation. Variation of gene expression in technical and biological replicates was low and did not explain the observed findings (Supplementary Fig. 1, and Orozco et al.[32]). Moreover, RNA deconvolution indicates that the peritoneal cavity was repopulated with inflammatory macrophages after thioglycollate treatment in all strains (Supplementary Fig. 2).

We analysed the gene expression levels of key polarization genes in human alveolar macrophages and found considerable variation among healthy volunteers under homoeostatic conditions (Fig. 1e, left column; an additional data set of 70 humans is shown in Supplementary Fig. 3). After LPS treatment, iNOS, arginase and IL-12β were variably upregulated among individuals. Strikingly, IL-12β showed a wide range of upregulation between 1.4-fold and 64-fold (Fig. 1e, bottom right). Similarly, using 26 classical inbred strains of the HMDP, we found a comparable diversity in peritoneal macrophage transcriptome data. At baseline, arginase expression showed a large variation across all strains, whereas iNOS and IL-12β were expressed at low levels (Fig. 1f, left column). After LPS treatment, the range of IL-12β upregulation between strains was large (0.4-fold to 64-fold). Other key macrophage genes such as MRC1 (CD206), MGL1, MGL2, FIZZ1 and IL-10 as well the housekeeping gene ACTB (β-actin) did not change in response to LPS (Supplementary Fig. 4). IL-6 was variably upregulated across the HMDP, and highly correlated with IL-12β (Supplementary Fig. 4). We confirmed expression differences between strains at the protein level showing great variation in IL-12 p70 levels in supernatants of LPS-stimulated peritoneal macrophages and varying iNOS and arginase induction patterns using intracellular flow cytometry (Supplementary Fig. 5). For the protein level data, we developed a representative panel of 13 strains that largely resemble gene expression profiles of the entire HDMP with regard to Arg1, iNOS and IL-12β transcription (Supplementary Fig. 5). Altogether, these data suggest that inter-individual differences in macrophage activation responses can be found both in human populations and strains of the HMDP.

**Global map of LPS responsiveness in 83 mouse strains**. Most bioinformatics tools are not designed to dissect a spectral distribution in a heterogeneous population. Analysis of commonly regulated genes after LPS treatment among all 83 strains fails (Supplementary Fig. 6), suggesting inter-strain variations of the macrophage LPS response. To overcome this problem, we established a gene expression based factor that represents the degree of LPS-induced polarization (polarization factor). Historically, the LPS-induced polarization was defined by the arginine metabolism: M2 is Arg1^{high} iNOS^{low}; M1 is Arg1^{low} iNOS^{high}[36]. We calculated polarization factors based on both iNOS and IL-12β. By dividing iNOS or IL-12β by arginase-1 gene expression values averaged to the mean of the HMDP (for details see Method section), 83 mouse strains were ranked to show their LPS responsiveness. We found that there is a continuous spectrum of LPS-induced activation strength; thus, identifying inherent LPS non/low- and -high responders (for example, FVB/NJ, CE/J and KK/HIJ, PL/J, respectively) (Supplementary Data 1). IL-12β- and iNOS-based rankings of mouse strains are highly correlated (Supplementary Fig. 7). In agreement with this ranking, analysis of amino acid levels in supernatant before and after LPS stimulation showed LPS high-responder strains producing less ornithine and more citrulline (Supplementary Fig. 8). Since the IL-12β based polarization factor has a higher resolution due to a more robust upregulation (Fig. 2c,d;

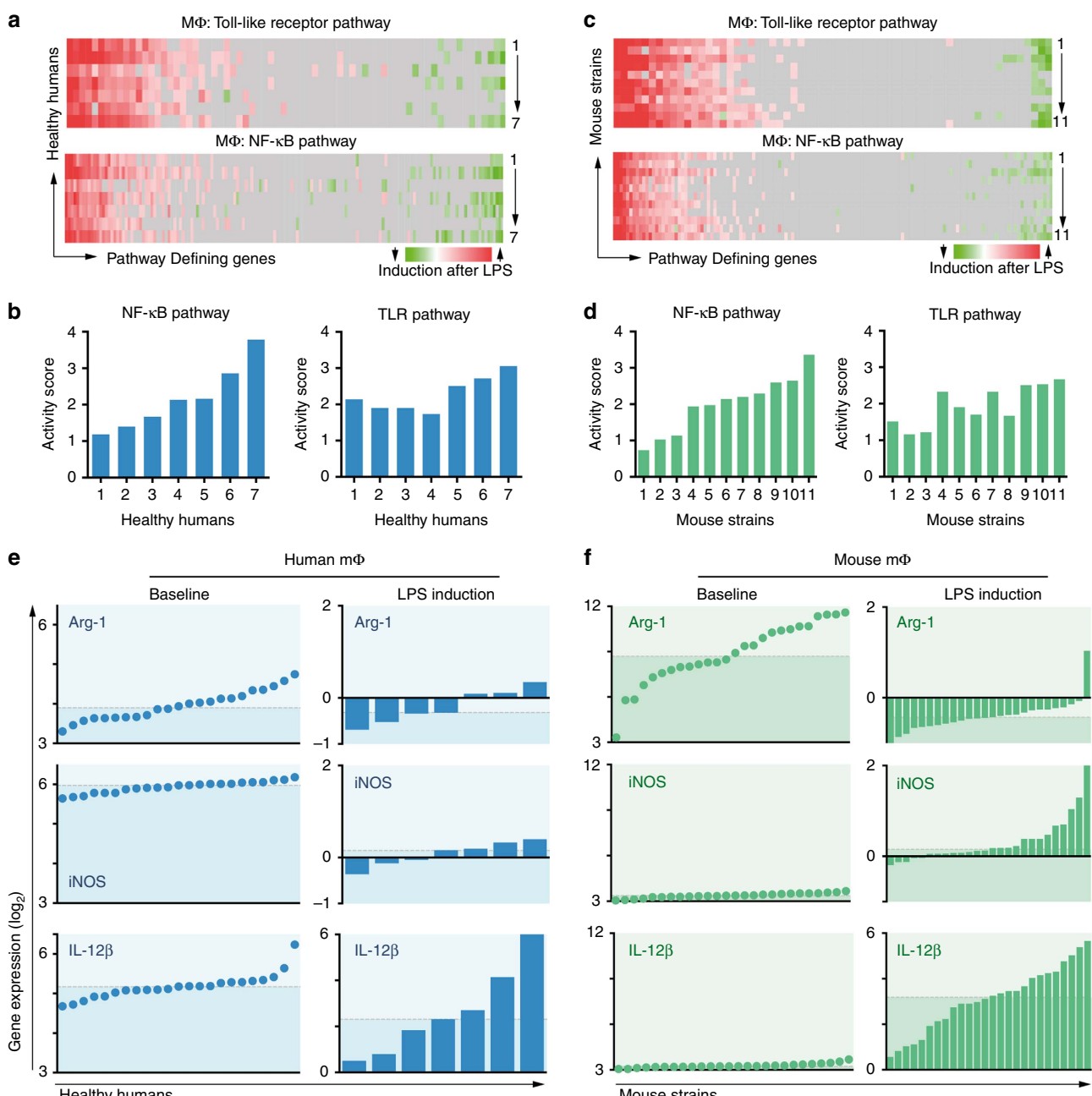

**Figure 1 | Natural variation of macrophage activation in response to LPS in humans and inbred mouse strains.** (**a**) Human alveolar macrophages were RNA sequenced after intrabronchial instillation of LPS or saline in seven healthy individuals. The fold change gene expression between LPS and control of key genes (columns) of the Toll-like receptor- (total 69 genes) and NF-κB signalling (total 155 genes) pathways are plotted for each volunteer (rows) using Ingenuity. Data derived from GSE40885. (**b**) Activity score ($z$ score) was calculated by Ingenuity based on matched and predicted induction of downstream gene expression after LPS treatment in the NF-κB and TLR pathway. Data sorted ascendingly by NF-κB activity score. (**c,d**) Thioglycollate-elicited peritoneal macrophages of 11 mouse strains of the hybrid mouse diversity panel were treated with LPS or PBS and their transcriptome was analysed by Affymetrix Array. Pathway analysis similar to **a** + **b**. (**e**) Arginase (Arg-1), iNOS (Nos-2) and IL-12β (p40) gene expression as RMA (robust multi-array average: quantile normalized, background-corrected, $\log_2$ transformed intensities) of human alveolar macrophages of 23 healthy human volunteers at baseline. The induction of gene expression ($\log_2$ fold change) after LPS treatment is shown in the bar chart. The dashed lines indicate the median. Baseline data from GSE27002, LPS-data from GSE40885. (**f**) Thioglycollate-elicited peritoneal macrophages (baseline and LPS-treated) of 26 classical recombinant mouse strains were analysed as described in **e**.

Supplementary Fig. 7) and iNOS relevance is controversial in human macrophages[26,27], the IL-12β based polarization factor was used in subsequent analyses.

**Gene signatures correlated with LPS responsiveness.** Next, genes of the peritoneal macrophage transcriptomes of all HMDP strains were correlated with the polarization factor, resulting in a ranked list of 1,276 LPS-positive responder genes and 2,619 LPS negative-responder genes at a false discovery rate (FDR) < 5% (Benjamini Hochberg) and $P < 0.0001$ (Pearson) (Fig. 3a,c; Supplementary Data 2). These lists were termed M(LPS)$^+$ and M(LPS)$^-$, according to a recent nomenclature proposal[20].

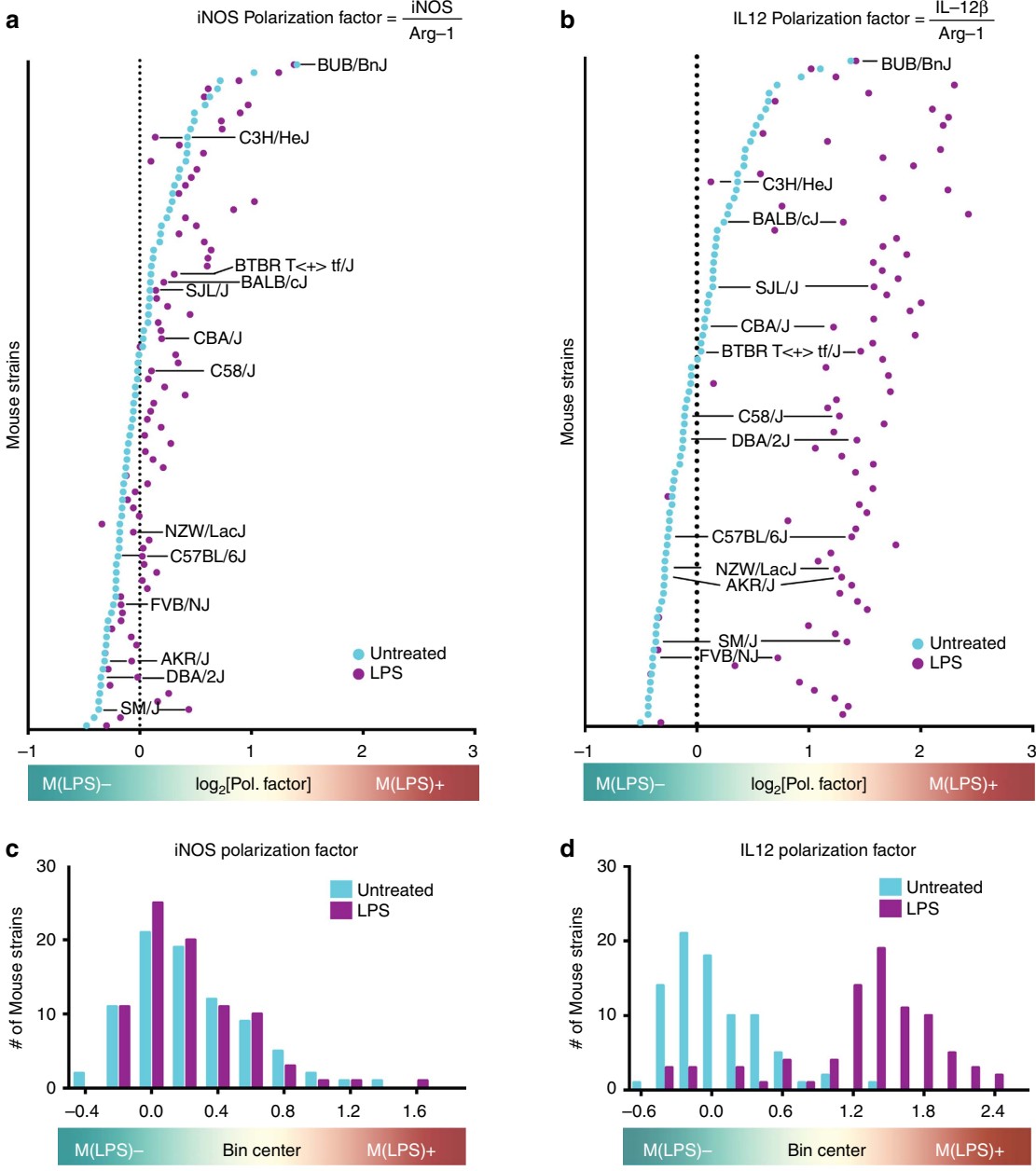

**Figure 2 | Inter-strain differences of LPS-induced macrophage activation across 83 mouse strains of the HMDP.** (**a,b**) Thioglycollate-elicited peritoneal macrophages were harvested from 83 mouse strains of the HMDP without (baseline, cyan) and with LPS (purple) treatment for 4 h *in vitro* and the genome-wide transcriptome was analysed. The polarization factor is based on the iNOS (**a,c**) or IL-12β (**b,d**) divided by Arg-1 gene expression ratio and determines the overall polarization state of each mouse strain in response to LPS (see Method section for details). The vertical dashed line indicates the median at baseline. Ranking details and strain names are provided in Supplementary Data 1. (**c,d**) Frequency histogram of LPS-induced macrophage polarization factor across 83 mouse strains at baseline (cyan) or after 4 h LPS (purple).

M(LPS)$^+$ genes were positively correlated with the polarization factor, for example, *IRF5* (ref. 37) and *VCAM1*, and M(LPS)$^-$ genes were negatively correlated, for example, *SUMO3* (ref. 38) and *NRD1* (Fig. 3b). We compared these lists with several published M1 macrophage gene signatures (LPS-treated; based on one mouse strain or pooled samples), and found only little overlap (Supplementary Fig. 9; Supplementary Table 1), suggesting that a population spectrum-based approach yields different results.

A functional classification based on Ingenuity's pathway categories showed enrichment for leukocyte migration, chemotaxis, inflammatory response and lymphocyte proliferation in M(LPS)$^+$ genes, whereas in M(LPS)$^-$ genes, apoptosis, DNA repair, cell cycle progression and metabolic genes were dominant (Fig. 3d). Gene lists contained 42 and 227 transcription factors in LPS positive-responders and negative-responders, respectively, providing a comprehensive map of the transcription factor landscape associated with LPS responsiveness. Known M1 transcription factors such as *IRF5*, *HIF1A*, *CEBPE* and *STAT6* were validated[39], and new candidates with high significance are proposed, for example, *TRIM24*, *KDM5A* and *TP53* (Supplementary Data 3; Fig. 3e).

**Activation states of murine tissue-resident macrophages.** The diversity of gene expression profiles of murine tissue-resident macrophages has been reported previously[19,40]. We extended this

data by determining the degree of activation at steady state using gene set enrichment analysis (GSEA)[41]. As expected, all tissue-resident macrophages are not predominantly M(LPS)$^+$ enriched.

However, they show differences in the strength of M(LPS)$^-$ gene expression (Fig. 4a). Microglia have the strongest M(LPS)$^-$ enrichment, whereas lung-resident macrophages also express

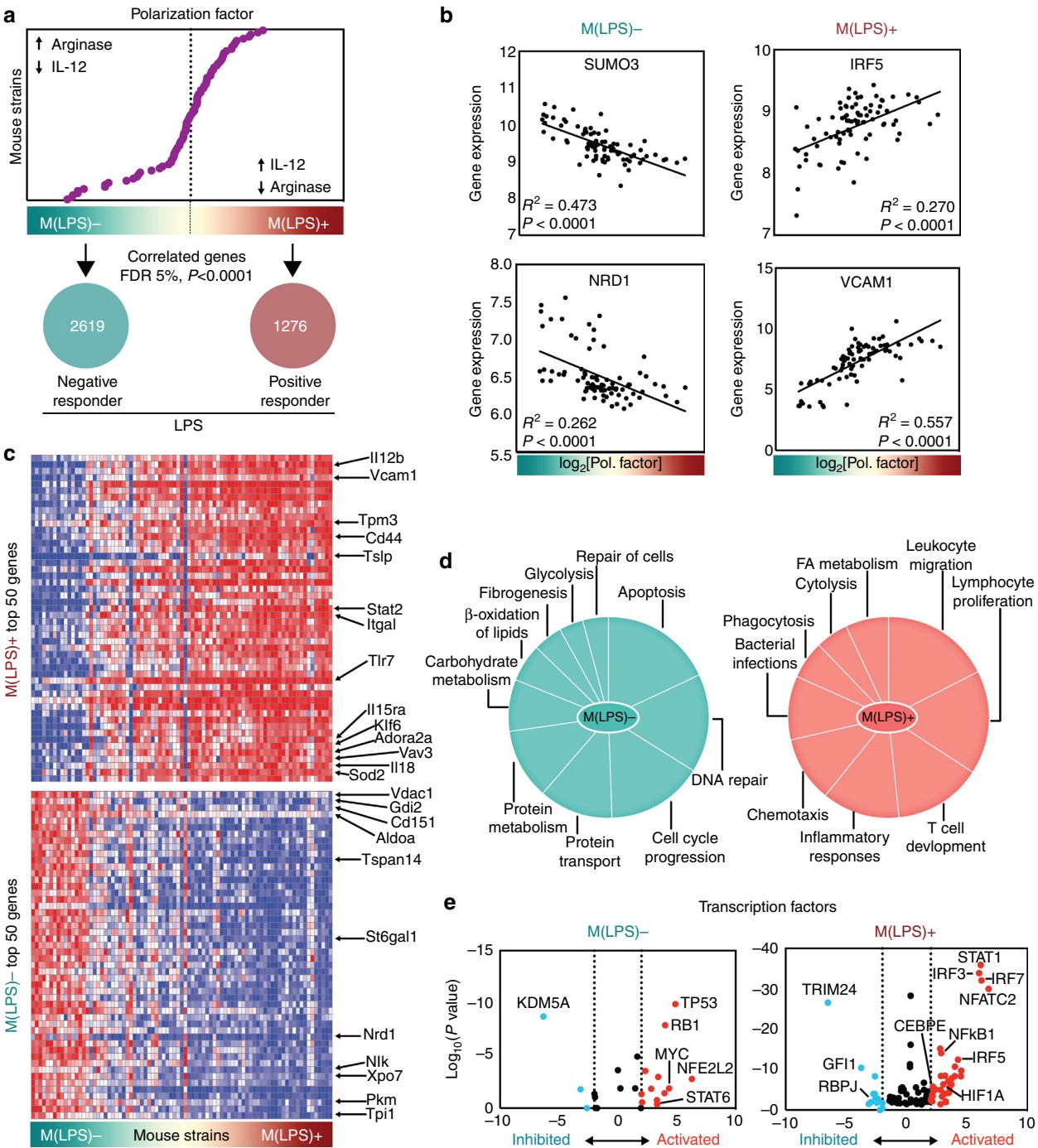

**Figure 3 | Core gene signatures and functional programs of LPS-responder and non-responder macrophages. (a)** Genome-wide transcriptome data for peritoneal macrophages of 83 strains of the HDMP was correlated with the IL-12β/Arg-1-based polarization factor (Fig. 2) with a FDR 5% (Benjamini Hochberg) cutoff and $P < 0.0001$ (Pearson). Positively and negatively correlated genes are designated M(LPS)$^+$ and M(LPS)$^-$, respectively, and indicate the responsiveness to LPS. Complete gene lists are provided in Supplementary Data 2. **(b)** Examples of genes that show a significant correlation with activation propensity. Each data point represents one mouse strain. Pearson's correlation between gene expression (RMA) and polarization factor (x axis), and significance indicated. **(c)** Heatmaps for top 50 M(LPS)$^+$ (top) and M(LPS)$^-$ (bottom) genes. Overall, 83 mouse strains of the HDMP are ordered on the x axis from low to high LPS responsiveness based on the polarization factor. Red = upregulation, blue = downregulation. Complete gene lists in Supplementary Data 2. **(d)** Key biological processes enriched in M(LPS)$^+$ and M(LPS)$^-$ gene signatures as determined by Ingenuity. The size of each segment indicates the significance of enrichment as $-\log_{10}$ ($P$ value). $P$ value cutoff of 0.001. **(e)** Transcription factors (TF) were extracted from M(LPS)$^{+/-}$ genes and their activation score was calculated using Ingenuity. The activity score (z score) indicates the level of consistently regulated (activated of inhibited) downstream genes. Activated and inhibited TFs are highlighted red and blue, respectively. The default cutoff for activation is $+/-$ 2 (dashed lines). A complete TF list is provided in Supplementary Data 3.

M(LPS)$^+$ genes (Fig. 4a,b). This likely reflects the natural exposure of lung-resident macrophages but not microglia to organisms of the commensal microbiota.

**Prediction of inflammatory disease in humans.** To apply our mouse findings to humans, we first tested whether the M(LPS)$^+$ and M(LPS)$^-$ gene lists sufficiently mapped to the human transcriptome. Human alveolar macrophages expressed about 70% of the mouse-derived gene lists (135 of the top 200 M(LPS)$^+$ genes; 142 of the top 200 M(LPS)$^-$ genes). As expected, resting human alveolar macrophages showed no significant M(LPS)$^+$ gene enrichment under baseline conditions and but a strong shift to M(LPS)$^+$ after LPS treatment (Fig. 4c; Supplementary Fig. 10). Similar enrichment was observed using human CD14$^+$ monocyte-derived macrophages with *Listeria monocytogenes* infection or mouse microglia with LPS challenge (Supplementary Fig. 10).

As polarized macrophages are known to critically shape pathophysiology of many inflammatory diseases[18], we tested

whether LPS responsiveness indicated by these gene signatures correlates with disease in humans. Peripheral blood leukocytes of systemic inflammatory response syndrome (SIRS) and sepsis patients both at day 1 and 3 after clinical diagnosis demonstrate gradually increasing M(LPS)$^+$ enrichment scores demonstrating sensitivity to infection severity (Fig. 4d). Isolated healthy human synovial macrophages were enriched in M(LPS)$^-$ genes, whereas macrophages from rheumatoid arthritis patients were strongly skewed towards an M(LPS)$^+$ phenotype (Fig. 4e). In lupus erythematodes, macrophages accumulate in the glomerula of the kidney and fuel disease progression[42]. Laser-dissected glomerula were M(LPS)$^+$ enriched in lupus nephritis, but not in healthy conditions (Fig. 4f). Importantly, in these data sets each individual patient showed varying degrees of gene enrichment. These findings suggest that the HMDP-derived gene signatures are applicable across species and in many tissues.

**Inter-individual activation state predicts tumour survival.** Tumour-associated macrophages (TAM) are of key importance in the tumour microenvironment that is known to reinforce M2

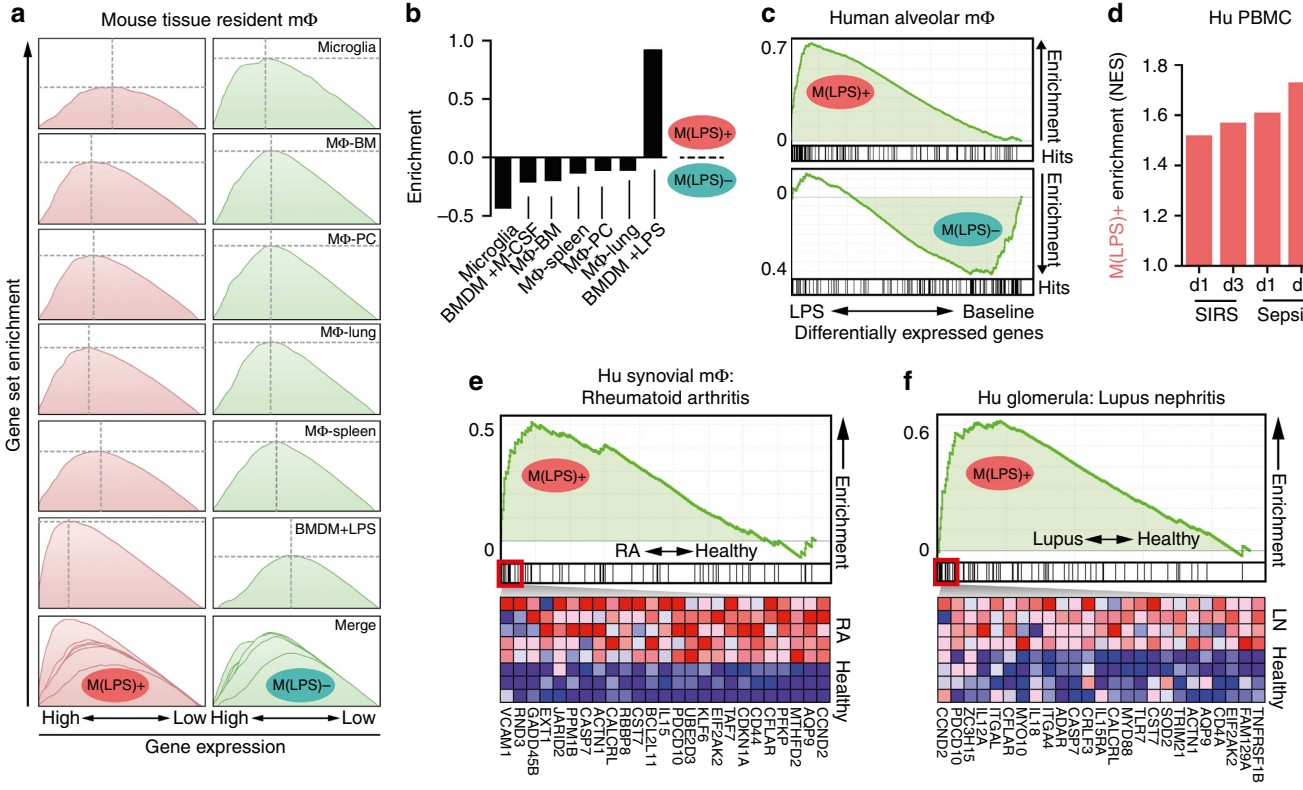

**Figure 4 | Macrophage activation phenotypes in mice and humans under homoeostatic and disease conditions. (a)** Homoeostatic tissue-resident macrophage populations of C57BL/6J mice were compared in their expression of the M(LPS)$^+$ and M(LPS)$^-$ signature using GSEA. Bone-marrow derived macrophages (BMDM) + LPS served as M1-skewed control. All data from the Immgen project (GSE15907 with $n = 2$–3 averaged per cell type) and BMDM + LPS from GSM1151665. **(b)** Classification of tissue-resident macrophages (C57BL/6J mice) in the M(LPS)$^{+/-}$ activation axis by dividing the GSEA normalized enrichment score (NES) for M(LPS)$^+$ and M(LPS)$^-$ genes for each data set. Numbers greater or smaller than 0 indicate high and low LPS responsiveness, respectively. BMDM-M-CSF from GSM1150712. Source of other data indicated in a. **(c)** GSEA of M(LPS)$^{+/-}$ signatures (top 200 genes) in human alveolar macrophages obtained by bronchoscopic lavage before (bottom panel) and after LPS (top panel) instillation. All genes of the control and LPS data set are ranked by differential expression. Vertical lines indicate a match of genes with the respective signature. Data from GSE40885. **(d)** Human isolated peripheral blood mononuclear cells (PBMC) were sequenced from patients presenting with systemic inflammatory response syndrome (SIRS) and sepsis at day 1 (d1) and 3 (d3) after clinical diagnosis. The transcriptome was analysed for enrichment of M(LPS)$^+$ genes in differentially expressed genes (disease versus healthy control) using GSEA. Data from GSE13904. **(e)** GSEA of M(LPS)$^+$ signature genes in isolated human synovial CD14 + macrophages from healthy ($n = 3$) and rheumatoid arthritis (RA) patients ($n = 5$). The top 25 leading edge genes are shown as heatmap (red = upregulated, blue = downregulated). Data from GSE10500. **(f)** GSEA of M(LPS)$^+$ signature genes in laser-dissected glomerula from healthy controls ($n = 14$) and patients with lupus nephritis ($n = 32$). The top 25 leading edge genes are shown as heatmap for four controls and four lupus patients (red = upregulated, blue = downregulated). Data from GSE32591.

and suppress M1 polarization[24,25,43]. Specific pharmaceutical modulation of polarization states has emerged as a new anti-cancer therapy[44,45]. The monocyte/macrophage content in solid tumours can exceed more than 50% of all leukocytes (Fig. 5a, Supplementary Fig. 11). Using RNA deconvolution, the $M(LPS)^{+/-}$ phenotypes are readily detectable in macrophage transcriptomes under controlled conditions (Supplementary Fig. 11). In the tumour microenvironment using bulk tumour biopsies, we demonstrate that most macrophages show enriched expression of $M(LPS)^-$ genes, which substantially varies between patients (Fig. 5b; Supplementary Fig. 11). To address the question whether $M(LPS)^+$ signatures can predict survival, we employed the PRECOG database that ranks genes by overall tumour

survival[46]. In collapsed data from 18,000 biopsies across 39 tumours, we find that the $M(LPS)^+$ signature correlates with survival, whereas the $M(LPS)^-$ signature correlates with cancer death (Fig. 5c). This pattern was found in many cancers of different ontologic origin, for example, osteosarcoma, melanoma, chronic lymphocytic leukaemia, Burkitt lymphoma and large-cell lung carcinoma (Supplementary Fig. 12). Patients with high expression of a disease-specific set of $M(LPS)^+$ or $M(LPS)^-$ genes show increased or decreased survival in multiple tumour entities, respectively (Fig. 5d–f). A comparison with several published M1 macrophage gene lists (Supplementary Fig. 9) indicates that survival prediction by $M(LPS)^{+/-}$ signatures is more robust in all data sets (Supplementary Fig. 13).

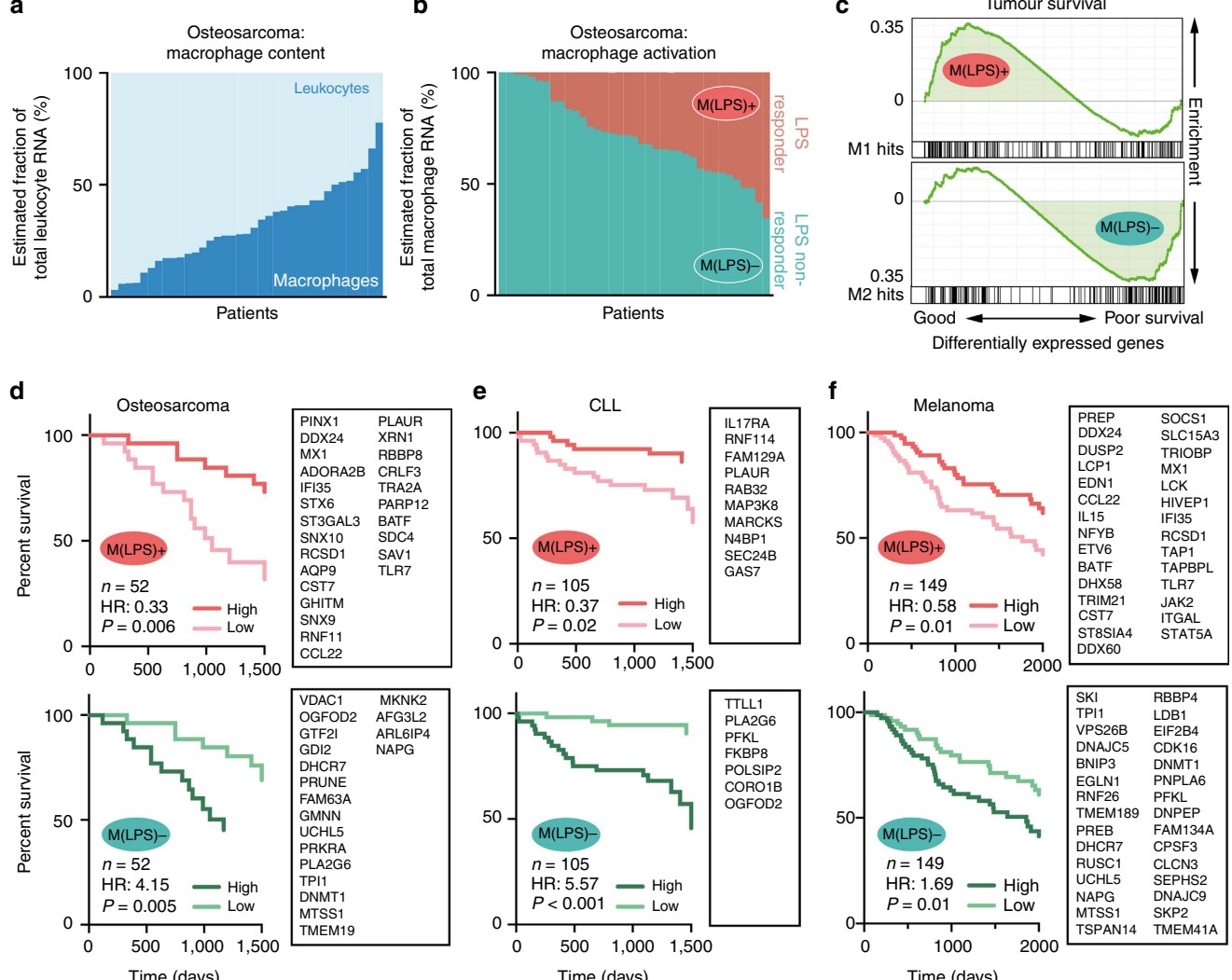

**Figure 5 | Activation state of tumour-associated macrophages can be detected in cancer biopsy transcriptomes and predicts survival. (a)** Percentage of macrophage-related RNA compared with total leukocyte RNA in raw osteosarcoma biopsies from 37 patients as detected by CIBERSORT deep deconvolution algorithm[59]. The groups labelled 'leukocytes' and 'macrophages' include gene signatures for lymphocytes, NK cells, granulocytes, dendritic cells and mast cells, and monocytes and macrophages, respectively. Data from GSE39055. **(b)** $M(LPS)^+$ and $M(LPS)^-$ gene signatures (top 200 genes) were applied on 37 raw osteosarcoma biopsies using CIBERSORT deconvolution[59]. Data from GSE39055. **(c)** Gene enrichment analysis of $M(LPS)^{+/-}$ gene signatures (top 200 genes) in the PRECOG data set (genes ranked by patients' survival in collapsed data of ≈18.000 tumour biopsies of 39 cancer entities[46]. **(d–f)** Survival data for human cancers was analysed for the impact of $M(LPS)^+$ (top) and $M(LPS)^-$ (bottom) gene expression in the tumour biopsy transcriptome. The study population was divided by the median of the mean expression of a tumour-specific gene list denoted in the adjacent boxes. For details see method section. Kaplan–Meier curves were plotted using Proggene[60]. Number of patients (*n*), hazard ratio (HR, cox proportional hazard analysis) and significance (log rank *P* value) are shown. Data from GSE21257 (Osteosarcoma), GSE22762 (Chronic lymphocytic leukemia) and SKCM-TCGA (Melanoma).

## Discussion

Here, we showed how a spectrum of macrophage phenotypes in many different inbred strains can be used to extract gene signatures of LPS responsiveness. Since the analysis of commonly LPS regulated genes in all strains failed, we established a surrogate marker based on *IL12β* and *Arg1* gene expression. Compared to published signatures, the resulting gene lists are unique, yet more robustly predictive of many human inflammatory and malignant disorders, suggesting that accounting for immune diversity in a heterogeneous population increases the translatability of mouse data.

The robust correlation of a large number of genes with the polarization factor across the HMDP reveals that the M(LPS) axis is a major player in macrophage biology. Out of 12,980 genes expressed, 1,276 (9.8%) and 2,619 (20.2%) are positively and negatively correlated with the IL-12b/Arg1 polarization factor, respectively. This is not surprising per se. However, it is surprising that only three M(LPS)$^+$ genes (Dpep1, Gkn2 and Hoxd4) are found in resting (thioglycollate-elicited) peritoneal macrophages, and only 11 M(LPS)$^-$ genes (*Arg1, Tpi1, Gpi1, Mif, Pkm, Zmynd8, Pcbp4, Myo6, Grk6, Egln1, Slc44a2*) suggesting that the LPS challenge exposes the activation propensity. Thus, it is more accurate to speak of individuals with an M1-skewed response to TLR challenge, rather than *a priori* M1-skewed individuals.

Inter-individual variability of the LPS response in healthy human PBMCs has been linked to profoundly different cytokine responses[47]. We speculate that mice and human macrophages evolved to either fight (LPS positive responders) or tolerate and heal (LPS negative responders) infections[48]. TLR polymorphisms[49] and differences in TLR signalling pathways are candidates for a genetic basis underlying M(LPS)$^+$ and M(LPS)$^-$ macrophage phenotypes. Of course, this is an oversimplification, because it is well known that macrophage polarizations also vary along the time axis of the infection or disturbance[18]. For example, myocardial infarctions are characterized by an initial abundance of M1 macrophages, followed by an M2-dominate healing phase[50,51]. Our data could implicate that M(LPS)$^+$ individuals may be susceptible to autoimmune diseases and M(LPS)$^-$ individuals may have unfavourable outcomes if afflicted with cancer. Similarly, individual differences in T helper cell differentiation in healthy individuals has been linked to disease susceptibility[29,52]. Multiple susceptibility genes and loci to infectious disease[12], autoimmunity[53,54] and cancer[55,56] have been described and our data suggest that inter-individual variation of macrophage activation propensity might be a confounding variable involved.

While the human immune system is shaped by both environmental factors and genetics[16], only the latter plays a role in laboratory inbred mice. It remains to be investigated what exact mechanisms account for the inter-strain differences. The HMDP consists of 29 classic parental inbred strains, and about 80 recombinant inbred strains (BXD, CXB, BXA/AXB and BXH panels) that are crosses between C57BL/6J, C3H/HeJ and DBA/2J[31]. The TLR4-insensitivity of the C3H/HeJ strain is therefore represented to variable degrees in its recombinant progeny. Inter-strain differences in the Interleukin-6/Interleukin-10/STAT3 pathway may also affect macrophage activation. After thioglycollate, inflammatory monocyte-derived macrophages dominate the cellular content of the peritoneal cavity in all mouse strains (Supplementary Fig. 2). Future studies on peritoneal and other tissue-resident macrophages in response to various stimuli will more fully characterize macrophage immune diversity in mice and humans. The polarization factor tool developed here will be useful for such studies.

A striking finding of this study is that the M(LPS)$^{+/-}$ signatures are robust enough to predict cancer survival or death from mixed-cell biopsy material, containing cancer, stromal and inflammatory cells. This makes these gene panels well suited for predictive tests in personalized medicine. Particularly, with new generation cancer treatments that manipulate tumour-associated macrophage polarization, new diagnostics are necessary to monitor treatment efficacy[43,44]. A simple multiplex PCR or RNA-sequencing run could harbour enormous predictive value, matching or exceeding the value of traditional biomarkers, for example, prostate specific antigen or BRCA1/2, or histo-pathological cancer staging. Of note, in contrast to an increasing number of disease-specific genetic tests that were developed by gene-outcome association statistics, for example, in breast cancer[57], we extracted genes in a 'bottom-up' approach centring around macrophage biology. Given the ubiquitous disease-relevant role of macrophages[18], this yielded transcriptomic signatures with predictive value in a number of different disease entities. Furthermore, in contrast to conventional flow cytometry- or PCR-based estimations of macrophage polarization in ordinal scale, the developed gene panels allow a gradual classification, thus enabling a population-based assessment in high resolution necessary for clinical applications.

Picking as few as 13 mouse strains is sufficient to qualitatively represent the diversity in macrophage activation responses to LPS. Re-evaluating the robustness of a new biological finding in a selected, representative panel of HMDP strains may be cost effective and prudent before embarking on a clinical drug development programme. Of note, all HMDP strains are fully inbred (homozygous at all loci) and commercially available[30,31], providing immediate access for research facilities and allowing reproducibility studies. Therefore, this approach shows great promise to improve the translation of immunological research findings from mice to humans.

In conclusion, our population-based approach yields (1) improved gene signatures with higher predictive power; (2) a methodological framework to extract meaningful signatures of data sets from heterogeneous populations, and demonstrates (3) the usefulness of the HMDP as a valuable surrogate for human diversity in translational research.

## Methods

**Mice.** Male mice 6–10 weeks of age were obtained from Jackson Laboratories (Bar Harbor, ME, USA) and housed in pathogen-free conditions on chow diet (Ralston-Purina Co, St. Louis, MO, USA). Details on the HMDP have been reported before[30] and data is accessible online (http://systems.genetics.ucla.edu/data). All mouse strains used in this study are listed in Supplementary Data 1. Experimental procedures were approved by the Institutional Care and Use Committee (IACUC) at the University of California, Los Angeles.

**Macrophage culture and activation.** Murine peritoneal macrophages were elicited with thioglycollate (BD, Sparks, MD, USA; same batch for all strains) for 4 days. Cells from up to 4 mice were pooled and plated at $4 \times 10^6$ cells ml$^{-1}$ in DMEM + 20% FBS + 1% streptomycin/penicillin in duplicates or triplicates. After overnight culture, adherent cells were selected (adherence assay for macrophage enrichment). RNA deconvolution shows two main populations of inflammatory macrophages (F4/80 +, MHCII − and F4/80 −, MHCII +) and a small population of neutrophils (Supplementary Fig. 2). Cells were incubated with 2 ng ml$^{-1}$ LPS (List Biological, Campbell, CA, USA) or control in DMEM + 1% FBS for 4 h before harvest. The viability of cultured macrophages was determined for some strains[32]. Briefly, macrophages were stained with 2 μM calcein AM (Molecular Probes) and the absorbance was measured at 530 nm. Viability was >90% with no difference between LPS-treated and untreated conditions[32]. Multiple replicates of some strains allowed determination of experimental variability[32].

**Gene expression profiling.** Total RNA was obtained using RNeasy columns (QIAGEN, Valencia, CA, USA) with DNA digest according to manufacturer's instructions and subsequently hybridized to Affymetrix HT MG-430A chip arrays. Chip signals were transformed to robust multichip average (RMA). Raw

microarray data for peritoneal macrophages of the HMDP has been published before[32] and is deposited under the NCBI GEO accession number GSE38705.

**Amino acid detection.** For 13 selected mouse strains, peritoneal macrophages were cultured in the presence of $2 \, ng \, ml^{-1}$ lipopolysaccharide in DMEM + 1% FBS media for 0, 4 or 24 h. Supernatant was collected and stored at $-80 \, °C$. Supernatant was analysed for amino acid concentration using a Hitachi L-8900 Amino Acid Analyzer.

**Flow cytometry and cytometric bead array.** After LPS stimulation for 0, 4 or 24 h as described above, macrophages were harvested after removal of supernatant and washing with PBS. For flow cytometry, cells were stained with viability dye (Ghost UV450 or Ghost Red710 Tonbo Bioscience, San Diego) using an intracellular staining protocol (IC fixation and permeabilization staining kit, eBioscience). Antibodies included iNOS (CXNFT, eBioscience catalogue no. 12-5920), CD11b (M1/70, eBioscience catalogue no. 25-0112), F4/80 (BM8, eBioscience catalogue no. 48-4801), Arginase-1 (polyclonal, R&D Systems catalogue no. IC5868F). All antibodies were used as 1:200 dilution. Macrophages were analysed by flow cytometry on an LSRII instrument using FACS Diva software (BD Bioscience) and analysed using FlowJo software (Tree Star, San Carlos, CA). The cytometric bead array (BD Bioscience) was used to determine cytokine concentrations in macrophage supernatant after LPS treatment according to manufacturers' instructions.

**Polarization factor and correlated genes.** The polarization factor ratio (PFR) describes the macrophage polarization state in a diverse population based on Affymetrix-RMA gene expression values. For each mouse strain, iNOS (Nos2) or IL-12β was divided by Arginase-1 (Arg1) and averaged to the population's baseline average (Equations 1 and 2). This was performed for both baseline and LPS-treated conditions. An increase or decrease indicates a shift to M(LPS)$^+$ or M(LPS)$^-$, respectively.

$$PFR_{iNOS} = \frac{iNOS}{Arg1} \times \frac{Arg1_{mean}}{iNOS_{mean}} \qquad (1)$$

Equation (1): PFR iNOS-based. Gene expression values as RMA are averaged to the mean of a heterogenous population. iNOS = Nos2 (inducible nitric oxide synthetase), Arg1 = Arginase − 1.

$$PFR_{IL12\beta} = \frac{IL12\beta}{Arg1} \times \frac{Arg1_{mean}}{IL12\beta_{mean}} \qquad (2)$$

Equation (2): PFR IL-12β-based. Gene expression values as RMA are averaged to the mean of a heterogenous population. IL12β = Interleukin − 12 beta, Arg1 = Arginase − 1.

Each transcript of the Affymetrix chip (total of 39,000 probe sets) was correlated to the PFR (ranking from M(LPS)$^-$ to M(LPS)$^+$) in 83 mouse strains. Positive and negative correlations with $P < 0.0001$ (Pearson) and FDR < 5% (Benjamini Hochberg) were selected as M(LPS)$^-$ and M(LPS)$^+$ gene signature lists, respectively (Supplementary Data 2).

**Gene signature analysis.** The gene set enrichment analysis (GSEA) tool[41] allows to investigate whether a list of genes (for example, signature) is represented in differentially expressed genes of two given conditions (for example, control versus disease). We used the GSEA tool embedded in the GenePattern 2.0 framework[58] with standard settings (weighted, 100 iterations). For enrichment in a single data set, the pre-weighted GSEA algorithm was used. Statistical parameters as standard GSEA output of the data sets used in this manuscript are: GSEA (Fig. 4c): Top: normalized enrichment score (NES) = 1.72 , FDR = 0.006 , $P = 0.004$, 135 genes matched. Bottom: NES = − 1.40, FDR = 0.192, $P = 0.108$, 142 genes matched. GSEA (Fig. 4d): SIRS d1: $P = 63$, FDR = 0.142, $n = 23$; SIRS d3: 0.067, FDR = 0.141, $n = 4$; sepsis d1: $P = 0.024$, FDR = 0.077, $n = 32$; sepsis d3: $P = 0.004$, FDR = 0.005, $n = 20$. GSEA (Fig. 4e): NES = 1.61, $P < 0.001$, FDR = 0.028, Top 100 M(LPS)$^+$ genes, out of which 62 genes matched. GSEA (Fig. 4f): NES = 1.45, $P = 0.03$, FDR = 0.04, Top 100 M(LPS)$^+$ genes, 73 genes matched.

Ingenuity's pathway analyzer (IPA, Qiagen) was used to analyse pathway enrichment in M(LPS) signatures. Transcription factors were extracted from these signatures and the $P$ value overlap and activation $z$ score was calculated according to pathway overlap and gene activity (inhibition versus activation) using IPA.

For bulk RNA deconvolution into cellular subsets, the CIBERSORT algorithm was used[59]. Core signatures included either the provided LM22 signature of several leukocyte subsets (for example, naive B cells, memory B cells, plasma cells, naive CD4 T cells, CD4 memory cells, follicular helper T cells, γδ T cells, NK cells, monocytes, macrophages, dendritic cells, mast cells, eosinophils, neutrophils) or the HMDP-derived gene lists (top 200 genes). In survival analyses, we performed gene set enrichment analysis on the PRECOG (PREdiction of Clinical Outcomes from Genomic Profiles) database that ranks genes by clinical survival either in a collapsed pan-cancer or in tumour-specific data set[46]. Top 10 or 30 leading edge genes of M(LPS)$^+$ or M(LPS)$^-$ lists were subsequently used to define a tumour-specific gene set that was used to retrospectively predict survival in published data

sets. Survival data was analysed using PROGgene2 (ref. 60) and plotted as median without sub-cohort division in a Kaplan–Meier format. Significance was calculated using the log rank test.

**Statistics.** Statistical analysis was performed using GraphPad Prism (GraphPad Software, San Diego). Affymetrix gene chip data was normalized using the robust multi-array average (RMA) method ($= \log_2$, background-corrected, quantile normalized). Correlation analyses were based on Pearson's correlation unless otherwise indicated. The threshold for significant M(LPS)$^+$ or M(LPS)$^-$ genes was set as $P < 0.0001$ (Pearson) and FDR < 5% (Benjamini Hochberg) for the correlation between gene expression (RMA) and PFR.

**Data availability.** The microarray data of the HMDP mouse strains are available in a public repository from the NCBI website under the accession number GSE38705. Other pre-published data sets referenced in this study can be found under the accession numbers GSE21257 and GSE39055 (Osteosarcoma), GSE22762 (Chronic lymphocytic leukemia), GSE11969 (lung large-cell carcinoma), GSE4475 (Burkitt lymphoma), GSE15907 (Immgen project), GSM1151665 (BMDM + LPS), GSM1150712 (BMDM + M-CSF), GSE10500 (rheumatoid arthritis), GSE32591 (lupus nephritis). The SKCM-TCGA (Melanoma) data set is available in a public repository from the cancer-genome atlas website (https://cancergenome.nih.gov). The authors declare that all the other data supporting the findings of this study are available within the article, its Supplementary Information files or from the corresponding author upon reasonable request.

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

## Acknowledgements

This project was funded by NIH R01HL115232 to K.L., NIH HL28481 and HL30568 to A.J.L. and Deutsche Forschungsgemeinschaft (DFG) BU3247/1-1 to K.B.

## Author contributions

K.B., E.E. and P.G. performed most experiments and analyses. A.B.P. and C.P. performed additional bioinformatics. E.E., G.T. and D.W. provided flow cytometry data. A.J.L. and C.D.M. provided resources and interpreted data. K.L. and K.B. designed the study, analysed data and wrote the manuscript.

## Additional information

**Competing interests:** The authors declare no competing financial interests.

