## [Peer Review File · Nature Communications]

Reviewers' comments:

Reviewer #1 (Remarks to the Author):

Buscher, Ehinger et al analyze activation induced gene expression in human lung macrophages, define their variation across seven individuals and then compare this range of variation to the range of variation found in a mouse panel from the hybrid mouse diversity panel, HMDP. The purpose as I understand it is to understand the contribution of genetic variation in explaining differences in macrophage polarization. The issue is of importance as human immune system variation is poorly understood and very relevant.

1. The purpose for using the HMDP panel is poorly explained. The diversity of LPS induced signatures are not related back to their genetic origin which is known for each strain in the HMDP panel. Instead the authors claim that this panel is representative of the variation seen across 7 (very few) human individuals.
2. This statement "Together, this data suggests that differences in macrophage polarization responses are an inherent trait of diverse human populations that can be captured by the HMDP" seems far fetched given that only 7 individual humans have been used as reference here.
3. Was the LPS-induced gene signature established using data averaged across all mouse strains? Isn't it possible that different gene signatures were elicited in different mouse strains and thus a mouse strain could be interpreted as a non-responder to LPS using this signature when in fact the mouse responds but in a qualitatively different manner?
4. After establishing the LPS-induced signature of genes, the authors spend the remaining manuscript analyzing this signature in public datasets correlating it to survival etc. This exercise becomes difficult to understand without first explaining better the unique aspects of this particular signature over previous LPS-induced gene expression signatures published.
5. The authors should compare their gene expression signature to the many other published LPS induced signatures, what are the genes included? Unique genes, previously not implied genes etc? again, what does the HMDP approach add here that is beneficial over the data generated in human cohorts directly?
6. When analyzing the presence of this signature in cancer tissues etc, how does the authors LPS induced signature compare to other LPS induced gene signatures? If there is no difference, then what has the authors added to the field?

Minor issue:

Variation of gene expression in technical and biological replicates was low and did not explain the observed findings (data not shown). It would be more convincing to show this important control data as a supplementary figure.

Reviewer #2 (Remarks to the Author):

In this manuscript, Buscher et al. demonstrate a wide gene expression range across mouse strains. The authors focus on some important macrophage activation markers to try and make their case. They extend their findings to some pathological scenarios. An interesting correlation with Buscher et al. was recently published by Netea and colleagues in a series of Cell papers showing essentially the same findings: human immune cells (in Netea's case, mainly human PBMCs) show very wide genetically-determined ranges in cytokine and chemokine expression. Furthermore, one could argue that the same phenomena extend even to single cells, which have the same genome (e.g. Shalek et al. Nature). Therefore, the broad take home message of Buscher

et al. is that there is a lot of variability from mouse strain to mouse strain, a fact that is well understood by experimentalists. The key question is whether Buscher et al. have revealed new mechanisms that cause such variability?

Major comments:

1. A major problem concerns the data derived from thioglycollate-elicited peritoneal macrophages. The authors have drawn large conclusions about the data in Fig. 1f, for example. As far as this reviewer understands the experimental setup, mice were injected with thioglycollate and macrophages harvested and then stimulated. Two major experimental issues could potentially confound the data interpretation. First, the embryonic GATA6-dependent macrophages do not seem to have been accounted for. Perhaps the authors assumed the resident macrophages would 'disappear' via the macrophage disappearance reaction. However, another possibility is that the variation (or part of the variation) in gene expression derives from underlying differences in the relative ratios of inflammatory macrophages vs. resident, as elegantly shown by Loke (Blood). Therefore, the authors need to quantify the cell populations to exclude the possibility that mice strains on one end of the spectrum are not due to a defect in inflammatory monocyte recruitment into the cavity (other possibilities exist, and the latter is one of several). Second, and even more concerning is the use of thioglycollate itself – basically the mice are injected with a TLR/NLR agonist soup. As arginase 1 is induced by LPS (Mori) via the IL-6/IL-10/Stat3 pathway (Murray), the authors have to account for (i) differences in TLR responsiveness by strain and (ii) differences in IL-6 and IL-10 production. It is entirely possible that much of the variability in arginase 1 expression could be accounted for by differences in IL-6 amounts or other soluble factors.

2. Mechanistic insight into the 'range' of gene expression is lacking. For example, it is remarkable that the authors failed to tackle some simple experiments such as incubating rested/naïve macrophages with cavity supernatant from different strains on the spectrum. Basically, the mechanism of the range of gene expression is not accounted for.

3. In past studies, several of the authors have argued for a dichotomous macrophage activation state they called 'kill' or 'repair and heal'. If this was true, there should be two broad clusters of macrophage gene expression patterns (with noise). Instead, a continuum was found, at least for the mRNAs tested, disproving the kill or repair theory. Thus, early work by one of the authors using SCID mice on two different backgrounds only captured two points on a larger continuum. These problems were not discussed.

4. The data presentation used is complex to understand. For example, in 1F, there are strains that have Arg-1 expression at $2e3$ to $2e12$. What accounts for this? If the authors purified resident peritoneal, inflammatory peritoneal and bone marrow-derived macrophages and stimulated the $2e3$ strain with IL-4, would Arg-1 be made to the same extent as the $2e12$ strain? The same argument can be made for all the data: because of the problems enumerated in point #1, too many variables are involved and have not been segregated.

5. Overall, the data presentation requires major revision and is confusing.

6. The data in figure 5 could be important, but requires more development. The authors are using human osteosarcoma samples (there is no human subjects section in the main text, so as osteosarcoma is a childhood cancer, one might assume these are from children). We already know that large numbers of macrophages are generally (but not always) correlated with poor outcomes in cancer. However, it is unclear what the data means. There is no histology (i.e. how many macrophages are in each sample?), no stratification of patients (have they been treated, untreated?). It's impossible to interpret anything about this data.

Other issues:

1. There is no author contribution section.
2. Introduction line 3. 'Many translational studies failed to confirm...' But many did, otherwise drug development would never have to go through pre-clinical testing. This statement is biased, and should be altered.
3. Introduction. 'In contrast, M2 macrophages metabolize arginine to ornithine...' is only partly true, as so do M1 macrophages.
4. Introduction. 'Cross-inhibition'. This statement is an oversimplification of an untested theory. The statement should be modified, as an experimental condition where two independent macrophage populations where one is iNOS-deficient, and the other Arg1 deficient, are mixed in different ratios and the metabolic flux through both populations has not been done.
5. Results. Human AMs. Purity, criteria for purity, viability and human subjects protection are not described.
6. The purity and sorting schemes for the cavity macrophages were not described (see point #1).

Reviewer #3 (Remarks to the Author):

The study entitled 'Natural variation of macrophage activation as disease-relevant phenotype predictive of inflammation and cancer survival' by Buscher et al., examines macrophage activation variability in mouse and humans. The premise of the paper is to understand the spectrum of macrophage activation and correlate it with susceptibility/resilience to inflammatory and malignant disorders. To confirm variation in macrophage activation and to identify disease-relevant gene signatures authors have used LPS for macrophage activation. For Gram-negative infectious diseases use of LPS is most relevant; however, for diseases such as rheumatoid arthritis and lupus, endogenous activation factors such as IFN- γ should be considered as a more relevant correlate. Further, authors have focused on Inter-strain differences in LPS induced macrophage activation by checking M1 and M2 macrophage markers (iNOS, Arg1 and IL-12). It would have been relevant to include alternative activation state of macrophages by using IL-4 and IL-13 and then showing inter-strain differences in macrophage activation to determine if the inverse correlate can be observed.

In the Introduction authors have discussed about iNOS upregulation and macrophage (Mouse and human) activation by LPS. In literature its well established that there are fundamental differences between macrophages from mice and humans about iNOS activity. It should be discussed in the introduction in detail to make it easy for readers to understand.

In general, to characterize more completely the polarization state of baseline and stimulated macrophages, authors should consider including data beyond Arg1, iNOS and IL-12 β for other commonly used markers such as Fizz-1, Mgl-1, Mgl-2, CD206, etc.

FIG1: the authors have identified natural variation of macrophage activation in response to LPS stimulation in macrophages from humans and inbred mouse strains. However, in humans the authors used alveolar macrophages whereas in mice they used thioglycollate-elicited peritoneal macrophages. Please clarify why mouse alveolar macrophages were not used for comparison.

On Page 4 in the first paragraph of the Results section in relation to FIG 1, the authors note, 'Variation of gene expression in technical and biological replicates was low and did not explain the observed findings?' Please include this information in the supplementary figures.

FIG 2: Please explain why the time point of 4 hours was selected. In numerous studies, induction of Arg1 by other stimuli such as IL-4 peak between 24 and 48 hours with a delay of induction in the first 12 hours. This this early induction specific to LPS? If a time course was indeed performed, please consider including this in the Supplementary Figures.

In FIG4 and FIG5, the authors' use of diseases correlates in interesting and a compelling correlate to the potential impact of macrophage polarization on human disease and health.

Mechanisms behind strain variation in macrophage activation is fascinating yet there is no mechanistic evidence exploring these differences. Whereas this is beyond the scope of this manuscript, the authors should address various possibilities in the discussion.

Reviewer 1

1. The purpose for using the HMDP panel is poorly explained. The diversity of LPS induced signatures are not related back to their genetic origin which is known for each strain in the HMDP panel. Instead the authors claim that this panel is representative of the variation seen across 7 (very few) human individuals.

Thank you for this comment. We did not make our approach clear enough and have now improved the writing accordingly. Natural immune variation and diversity in immune reactions is a crucial feature of human populations (Netea MG, 2016). Yet, this variation is neglected in most mouse studies where usually only one strain is investigated. Therefore, we asked whether a mouse panel of inbred strains could mimic the dynamic range of variation in humans, and how this can be used to extract insightful gene signatures in translational research.

We agree that it would be interesting to study the genetic and molecular origin of this macrophage diversity, but this is beyond the scope of the present study. Here, we focus on deriving robust gene signatures using macrophage immune diversity in 83 mouse strains.

Analyzing natural variation within a population is not trivial, because bioinformatics tools for omics data are not designed to analyze a spectral distribution. Our methodological approach is different from other studies, where condition A (untreated) is compared to condition B (LPS treated) in one mouse strain or pooled samples. In fact, analysis of the up- and downregulated genes in response to LPS in each strain, and then selection of the commonly regulated genes across all strains fails (results in 0 genes; see new supplemental figure 6). In our study, we used an activation ratio (IL12b/ARG1) across all strains as a basis to extract positively (M(LPS)+) and negatively (M(LPS)-) correlating genes. As shown below (point 5), our signatures have little or no overlap with known LPS signatures and robustly predict tumor survival, where other LPS signature are much less robust (point 6).

We agree with the reviewer regarding the small sample size. We chose this small dataset with 7 individuals because of its high quality (baseline and LPS-treated sample from the same healthy individual (paired), and LPS was injected in vivo via bronchoscopy = fewer confounders than in other data sets). We additionally analyzed another data set with 70 humans that supports our conclusions (Review Figure 1).

Review Figure 1: Variation of gene expression in human alveolar macrophages from different donors. Data includes 70 humans at steady state. This data corresponds to manuscript figure 1. The ACTB gene is shown as control. Data from GSE13896.

We are not the first to introduce the concept of macrophage variation among humans. This has been already established and several reviews highlight the relation between macrophage phenotype and interindividual disease susceptibility (e.g. due to TLR polymorphisms, see review by Netea MG, 2012). Therefore, our data (Manuscript Figure 1) confirm that known key macrophage genes do vary between humans (due to many reasons), and this is also the case in different strains of the hybrid mouse diversity panel (due to genetics). This information is important to explain the use of IL12b and ARG1 in downstream analyses. We added the additional data sets in supplemental figure 3 and explained the purpose of this comparison in more detail.

2. This statement “Together, this data suggests that differences in macrophage polarization responses are an inherent trait of diverse human populations that can be captured by the HMDP” seems far fetched given that only 7 individual humans have been used as reference here.

We analyzed another published human macrophage transcriptome dataset with 70 individuals (alveolar macrophages) to measure the inter-individual variation of selected genes. As shown in Review Figure 1 this data also support diversity in key macrophage genes. We agree that future work should use a broader approach covering different macrophages in different organs to generally conclude about macrophage diversity in humans. We therefore rephrased this statement.

3. Was the LPS-induced gene signature established using data averaged across all mouse strains? Isn't it possible that different gene signatures were elicited in different mouse strains and thus a mouse strain could be interpret as a non-responder to LPS using this signature when in fact the mouse responds but in a qualitatively different manner?

Thank you for this comment. The LPS-induced gene signature was not established using data averaged across all mouse strains. We apologize for our misleading description of the M(LPS)+ and M(LPS)- signatures. M(LPS)+ genes are positively correlated with the IL-12/Arg1 ratio. M(LPS)- genes are negatively correlated. In some mice, the M(LPS)+ genes are highly expressed and in other mice the M(LPS)- genes. Yes, the response to LPS is indeed qualitatively different.

Conventional bioinformatics (differential expression analysis, Venn diagrams, clustered heatmaps) do not allow to study a gradient of responses in large populations. In fact, analysis of the up- and downregulated genes in response to LPS in each strain, and then selection of the commonly regulated genes across all strains fails (results in 0 genes; see new supplemental figure 6). This reflects the fact, as the reviewer pointed out, that each strain does not react to LPS in the same quantitative and qualitative manner, and the more strains are included, the less genes overlap. A core element of our work was therefore to establish a workflow that accounts for the differential gene responses. We found that a surrogate parameter, called here polarization factor (IL-12b/Arg1 gene expression), captures the dynamic range of responses in the mouse diversity panel. IL-12 and arginase are known key genes in the macrophage LPS response. A correlation matrix of the polarization factor and all genes across all strains identified genes that recapitulate the polarization factor response. Here, including all 83 strains, we found 2619 and 1276 genes with significant positive and negative correlation to the polarization factor, respectively. The signatures are unique and outperform known macrophage LPS response gene lists (see points 5 + 6). This shows that our approach is feasible to study spectral distribution across populations.

We modified the manuscript in order to point the reader more clearly to these novel aspects of analysis.

4. After establishing the LPS-induced signature of genes, the authors spend the remaining manuscript analyzing this signature in public datasets correlating it to survival etc. This exercise becomes difficult to understand without first explaining better the unique aspects of this particular signature over previous LPS-induced gene expression signatures published.

Please see response to point 5.

5. The authors should compare their gene expression signature to the many other published LPS induced signatures, what are the genes included? Unique genes, previously not implied genes etc? again, what does the HMDP approach add here that is beneficial over the data generated in human cohorts directly?

We thank the reviewer for this suggestion. We compared our gene signatures with several published ones. We picked the following data sets: one with a largely similar protocol (thioglycollate-elicited mouse macrophages), one with murine bone-marrow derived macrophages and one human monocyte-derived macrophage dataset. Details of the three signatures are shown in Table 1. These were compared to the top 200 M(LPS)+/- genes that we used in manuscript figures 3-5.

Name	Protocol	Number of genes	Source
Signature 1	C57BL/6 peritoneal macs thioglycollate i.p. 5d - LPS 6h in vitro (M1) - no M2 sig. available	M1: 200 genes	Schroder K, PNAS 2012
Signature 2	mouse bone-marrow derived macs - LPS + IFN γ 24h (M1) - IL-4 (M2)	M1: 57 genes M2: 33 genes	Jablonski K, Plos One 2015
Signature 3	human CD14+ monocyte-derived macs M-CSF polarized - LPS + IFN γ or TNF α (M1) - IL-4 or IL-13 (M2) (integrated from three different datasets)	M1: 106 genes M2: 58 genes	Becker M, Sci Rep 2015

Review Table 1: Experimental details of published macrophage polarization signatures[^].

Review Figure 2: Overlap between published macrophage activation gene signatures and our LPS-responder/non-responder (M(LPS)+/-) gene lists. a) Signature 1 b) Signature 2 c) Signature 3 (see details in table 1). In Signature 1 there is no M2 gene set available.

As shown in Review Figure 2, there is only little overlap overall between our and published signatures. This includes signature 1 that was also derived from LPS-treated thioglycollate-elicited macrophages. Some genes are present in the M(LPS+)-responder list (about 10-15%). That published M2 signatures do not overlap with the M(LPS)- lists is not surprising, as the former were derived from IL-4/IL-13-treated macrophages. These data suggest that a population-based approach yields significantly different results than data derived from pooled mouse or human macrophages treated with cytokines. This data is now shown in the manuscript supplemental figure 9.

6. When analyzing the presence of this signature in cancer tissues etc, how does the authors LPS induced signature compare to other LPS induced gene signatures? If there is no difference, then what has the authors added to the field?

We thank the reviewer for this suggestion. After having established that our approach results in different gene signatures compared to published ones (see point 5), we now directly compare the performance of these signatures.

Review Figure 3: Survival analysis using published gene signatures. Signatures 1-3 are described in Table 1. Hazard ratio and p-value are indicated. Red = high, green = low signature expression. This figure corresponds to manuscript figure 5 (shown below), where the same datasets were analyzed using $M(LPS)^+$ and $M(LPS)^-$ signatures. Data from GSE21257 (Osteosarcoma), GSE22762 (Chronic lymphocytic leukemia), and SKCM-TCGA (Melanoma).

Review figure 4: This figure is copied from the manuscript (figure 5) and shows the performance of the *M(LPS)+* and *M(LPS)-* gene signatures on the same datasets used in Review figure 3.

As shown in Review Figure 3, the published signatures detect a significant survival difference in only some tumor datasets. Signature 1, which was derived from B/6 mouse peritoneal macrophages (one strain) treated with LPS, successfully resolves only the CLL dataset but not melanoma or osteosarcoma. M1 signatures tend to mark a subpopulation of patients with lower mortality, similar to our data. In contrast, our signatures work in all datasets (Review figure 4; Manuscript figure 5). We added these panels in the manuscript supplements (suppl. fig. 13). These data suggest that a population-based approach to extract gene signatures seems more robust and sensitive when applied to heterogeneous human cohorts.

Therefore, we believe our work adds three main points to the field: 1) Gene signatures can be improved by accounting for immune diversity. 2) A methodological framework to extract meaningful signatures of datasets in large heterogeneous populations. 3) The hybrid mouse diversity panel could be a valuable surrogate for human diversity in translational research.

Minor issue:

Variation of gene expression in technical and biological replicates was low and did not explain the observed findings (data not shown). It would be more convincing to show this important control data as a supplementary figure.

These quality controls were already published (Orozco LD, 2012) and are reproduced below (Review figure 5). Pathway activation in biological replicates are shown in Review figure 6. These data are now shown in supplemental figure 1.

Review figure 5: Biological replicates of peritoneal macrophage transcriptomes cluster mostly together (Data from Orozco, Lusic et al, 2012).

Review figure 6: NF-kB pathway activation of biological replicates. Related to figure 1. a,b) The NF-kB pathway activation was determined in macrophage transcriptomes by IPA using biological replicates of classical mouse strains.

Reviewer #2 (Remarks to the Author):

Major comments:

1. A major problem concerns the data derived from thioglycollate-elicited peritoneal macrophages. The authors have drawn large conclusions about the data in Fig. 1f, for example. As far as this reviewer understands the experimental setup, mice were injected with thioglycollate and macrophages harvested and then stimulated. Two major experimental issues could potentially confound the data interpretation. First, the embryonic GATA6-dependent macrophages do not seem to have been accounted for. Perhaps the authors assumed the resident macrophages would ‘disappear’ via the macrophage disappearance reaction. However, another possibility is that the variation (or part of the variation) in gene expression derives from underlying differences in the relative ratios of inflammatory macrophages vs. resident, as elegantly shown by Loke (Blood). Therefore, the authors need to quantify the cell populations to exclude the possibility that mice strains on one end of the spectrum are not do to a defect in inflammatory monocyte recruitment into the cavity (other possibilities exist, and the latter is one of several).

We thank the reviewer for raising this concern, which we had not addressed sufficiently in the original submission. It has been shown that GATA6+ tissue-resident macrophages, which make up the large peritoneal macrophage population, disappear after thioglycollate injection (Okabe & Medzhitov, 2014). We agree with the reviewer that different patterns of invading macrophage types in different strains could confound our data. To address this, we determined the fractions of inflammatory cell populations in the transcriptome of all mouse strains by RNA deconvolution. We used the reference signatures of the Immgen consortium database. Here, different steady-state tissue resident and thioglycollate-elicited macrophage populations were sorted and sequenced. We deconvolved the full transcriptomes of all mouse strains with Cibersort (Newman AM, Nat Methods 2015). This allows estimating the cellular composition in a convoluted transcriptome based on signature gene expression profiles.

Review Figure 1: RNA deconvolution of the full transcriptome of the hybrid mouse diversity panel (83 strains) using Immgen reference signatures (GSE15907) of different cell types in the peritoneal cavity (PC). Blue = neutrophils in the thioglycollate-treated PC. Orange = large peritoneal macrophages in the untreated PC were not detected. Green and red = two macrophage populations in the thioglycollate treated PC. Thioglycollate treatment for 5 days. PM = peritoneal macrophages.

We now show that > 90% of peritoneal cells bear the gene signature of thioglycolate-induced macrophages in all strains, while peritoneal neutrophils accounted only for a small fraction (Review Figure 1). We did not observe large differences between the strains, suggesting that i.p. thioglycollate for 5 days had a similar effect in all strains. The gene signature of large peritoneal macrophages was not detectable after thioglycolate treatment, indicating a complete repopulation of the peritoneal cavity with inflammatory macrophages as shown before (Okabe 2014). These data are now shown in supplemental figure 2.

2. Second, and even more concerning if the use of thioglycollate itself – basically the mice are injected with a TLR/NLR agonist soup. As arginase 1 is induced by LPS (Mori) via the IL-6/IL-10/Stat3 pathway (Murray), the authors have to account for (i) differences in TLR responsiveness by strain and (ii) differences in IL-6 and IL-10 production. It is entirely possible that much of the variability in arginase 1 expression could be accounted for by differences in IL-6 amounts or other soluble factors.

We thank the reviewer for this comment. We would like to clarify that it was not our intent to elucidate the reasons why different mouse strains show different macrophage attributes. We agree that it is entirely possible that different mechanisms including IL6/IL10/Stat3 and different responses to LPS could lead to different or common macrophage phenotypes in these strains. We are using the mouse diversity panel to mimic a broad range of reactions to an environmental stimulus (here LPS), similar to differential LPS responses in humans due to TLR polymorphisms (review Netea MG 2012).

The novelty of our work is that we take natural immune diversity into account to derive insightful gene signatures, and do not rely on one single inbred mouse strain like most studies do. This required a new bioinformatics approach because conventional tools (clustered heatmaps, differential expression) do not allow investigating a spectrum in large heterogeneous populations. In fact, analysis of the up- and downregulated genes in response to LPS in each strain, and then selection of the commonly regulated genes across all strains fails (results in 0 genes; see new supplemental figure 6). The goal of our study was therefore to investigate how meaningful gene signatures can be derived from omics data of a diverse population, and whether this yields insightful gene signatures that can robustly predict disease. We added data in the supplements that show that our signatures, compared to published ones, are mostly unique (little overlap), and outperform previous gene signatures in survival analyses (see new supplemental figure 9).

Differences in Arg1 expression could indeed be related to different IL-6 levels. Gene expression values of IL-10 and IL-6 at baseline and in response to LPS are depicted in Review Figure 2. Similar to IL-12 expression, IL-6 induction is very variable across mouse strains after LPS treatment. As control, the LPS-insensitive C3H mouse strain does not respond to LPS. IL-10 transcription is not detectable across strains. IL-6 induction highly correlates with IL-12b ($R^2 = 0.7$, $p < 0.001$) (Review Figure 3). There is no significant correlation of Arg1 and IL6 (Review Figure 3).

Review Figure 2: Gene expression of IL-6 and IL-10 in classical inbred strains of the hybrid mouse diversity panel showing a broad range of activation in LPS-treated macrophages for IL-6, but not IL-10.

Review Figure 3: Correlation of IL12b or Arg1 and IL6 gene expression in LPS-treated macrophages of different classical mouse strains. Linear regression. Pearson correlation, R^2 and p -value indicated. The correlation between IL6 versus Arg1 is not significant.

3. Mechanistic insight into the ‘range’ of gene expression is lacking. For example, it is remarkable that the authors failed to tackle some simple experiments such as incubating rested/naïve macrophages with cavity supernatant from different strains on the spectrum. Basically, the mechanism of the range of gene expression is not accounted for.

We thank the reviewer for this comment. We would like to clarify that it was not our intent to elucidate the reasons why different mouse strains show different macrophage attributes. We agree with the reviewer that it is entirely possible that different mechanisms including IL6/IL10/Stat3 and different responses to LPS could lead to different or common macrophage phenotypes in these strains. We are using the mouse diversity panel to mimic a broad range of reactions to an environmental stimulus (here LPS). Review figure 1 suggests that the cellular composition in the thioglycolate-treated peritoneal cavity is largely similar in all strains. While the proposed experiment is intriguing, this is beyond the scope of this work.

4. In past studies, several of the authors have argued for a dichotomous macrophage activation state they called 'kill' or 'repair and heal'. If this was true, there should be two broad clusters of macrophage gene expression patterns (with noise). Instead, a continuum was found, at least for the mRNAs tested, disproving the kill or repair theory. Thus, early work by one of the authors using SCID mice on two different backgrounds only captured two points on a larger continuum. These problems were not discussed.

We thank the reviewer for pointing this out. The dichotomous view of macrophage activation stems from experiments 20 years ago, when multiplexed technologies were not available. The M1 (kill)/M2 (heal) axis is contained in newer concepts of a spectrum of macrophage activation. Newer research has shown that the macrophage phenotype is strongly shaped by the environment, including infectious organisms, tissue of residence and cytokines (Perdiguerro & Geissmann, 2016). Combinations of these stimuli can elicit a broad spectrum of responses (see review by Murray 2014). Here, we specifically focus on mouse strain differences in macrophage response to LPS.

The phylogeny of the hybrid mouse diversity panel used in our study is relevant to understand population diversity. It consists of 29 classic parental inbred strains, and about 80 recombinant inbred strains (BXD, CXB, BXA/AXB, and BXH panels). The latter are derived from crosses between C57BL/6J, C3H/HeJ, and DBA/2J (Lusis AJ, 2016). The well studied TLR4-insensitivity of the C3H/HeJ strain is therefore represented to variable degrees in its derived recombinant strains. These implications have been laid out in the discussion.

5. The data presentation used is complex to understand. For example, in 1F, there are strains that have Arg-1 expression at 2e3 to 2e12. What accounts for this? If the authors purified resident peritoneal, inflammatory peritoneal and bone marrow-derived macrophages and stimulated the 2e3 strain with IL-4, would Arg-1 be made to the same extent as the 2e12 strain? The same argument can be made for all the data: because of the problems enumerated in point #1, too many variables are involved and have not been segregated.

We would like to clarify that it was not our intent to elucidate the reasons why different mouse strains show different macrophage attributes. We agree with the reviewer that it is entirely possible that different mechanisms including IL6/IL10/Stat3 and different responses to LPS could lead to different or common macrophage phenotypes in these strains. We are using the mouse diversity panel to mimic a broad range of reactions to an environmental stimulus (here LPS). Review figure 1 suggests that the cellular composition in the thioglycolate-treated peritoneal cavity is largely similar in all strains. It would be interesting in follow-up studies to investigate the dynamic range of macrophage phenotypes in different tissues.

6. Overall, the data presentation requires major revision and is confusing.

We changed parts of the manuscript to more clearly show the aim, novelty and the conclusions of this study.

7. The data in figure 5 could be important, but requires more development. The authors are using human osteosarcoma samples (there is no human subjects section in the main text, so as osteosarcoma is a childhood cancer, one might assume these are from children). We already know that large numbers of macrophages are generally (but not always) correlated with poor outcomes in cancer. However, it is unclear what the data means. There is no histology (i.e. how many macrophages are in each sample?), no stratification of patients (have they been treated, untreated?). It's impossible to interpret anything about this data.

We thank the reviewer for pointing this out. Our study did not include the collection and processing of human cancer samples. Instead, we used published datasets to assess the significance of the derived gene signatures. This approach allowed us to not only screen for one cancer type, but five different malignancies (osteosarcoma, CLL, melanoma, lung LLC, Burkitt lymphoma; see manuscript figure 5, and supplemental fig. 12).

The role of polarized macrophages in the tumor microenvironment has been studied before (Biswas & Mantovani, 2010). We are not claiming that the impact of invading macrophages on tumor survival is new. Instead, we use this well studied phenomenon and ask whether our gene signatures of LPS responsiveness can detect the phenotype of tumor associated macrophages. Histology does not provide this information.

Our gene signatures allow a quantitative measure of LPS responsiveness, thus identifying tumor biopsies in which macrophage genes that positively (M(LPS)+) correlate with IL12/Arg1 are over-represented. These responders show a favorable survival across all cancer types studied. The opposite is true for M(LPS)- enriched biopsies. Importantly, our gene signatures are unique (e.g. do not significantly overlap with published gene lists), and outperform them in survival analysis (data now shown in supplemental figures 9 and 13).

The source of all cancer datasets used is indicated in the manuscript. The databases contain clinical information about the cohorts studied. For example, the osteosarcoma data is available at EuroBoNet, and includes pre-chemotherapy samples in children. The corresponding publication shows a correlation of macrophage infiltration and tumor survival using histological assessment (Buddingh EP, Clin Cancer Res, 2011). We added more information of the used datasets in the methods section.

Other issues:

1. There is no author contribution section.

The author contributions can be found on page 18 (acknowledgments).

2. Introduction line 3. 'Many translational studies failed to confirm...' But many did, otherwise drug development would never have to go through pre-clinical testing. This statement is biased, and should be altered.

A series of remarks have been published that address the problem of failed translational studies (Hayden EC, Nature News, 03/2014; Reardon S, Nature News 02/2016; Perrin S, Nature News, 03/2014; Arrowsmith J, Nat Rev Drug Discov 2011). We have changed that sentence to a more general comment.

3. Introduction. 'In contrast, M2 macrophages metabolize arginine to ornithine...' is only partly true, as so do M1 macrophages.

We agree with the reviewer and changed that statement.

4. Introduction. 'Cross-inhibition'. This statement is an oversimplification of an untested theory. The statement should be modified, as an experimental condition where two independent macrophage populations where one is iNOS-deficient, and the other Arg1 deficient, are mixed in different ratios and the metabolic flux through both populations has not been done.

We agree with the reviewer. We deleted that sentence.

5. Results. Human AMs. Purity, criteria for purity, viability and human subjects protection are not described.

We did not conduct a human study, but used published datasets to investigate significance of our derived gene signatures. For all datasets used, the GSE accession numbers are indicated. The criteria asked for (human subject protection etc.) are outlined in the original publication of each data set (see references in the manuscript).

6. The purity and sorting schemes for the cavity macrophages were not described (see point #1).

We added this information in the methods section.

Reviewer #3 (Remarks to the Author):

It would have been relevant to include alternative activation state of macrophages by using IL-4 and IL-13 and then showing inter-strain differences in macrophage activation to determine if the inverse correlate can be observed.

We agree with the reviewer that this would be an interesting extension of the present study. Due to extensive labor and costs we so far only have LPS data for all 83 mouse strains. Other stimuli need to be addressed in the future.

In the Introduction authors have discussed about iNOS upregulation and macrophage (Mouse and human) activation by LPS. In literature its well established that there are fundamental differences between macrophages from mice and humans about iNOS activity. It should be discussed in the introduction in detail to make it easy for readers to understand.

We thank the reviewer for this remark. We changed introduction to emphasize this species difference. Note that we chose IL-12 and not iNOS for our polarization factor.

In general, to characterize more completely the polarization state of baseline and stimulated macrophages, authors should consider including data beyond Arg1, iNOS and IL-12 β for other commonly used markers such as Fizz-1, Mgl-1, Mgl-2, CD206, etc.

We show this data in Review Figure 1. Fizz1, MGL1, MGL2 and CD206, known markers for alternative activation, are not induced after LPS treatment. Interestingly, MGL1 (CD301) shows a genetic effect across the strains, similar to Arginase 1. We also looked at two key cytokines: Interleukin-6 transcription after LPS treatment is strongly upregulated with great variation between the strains. IL-6 highly correlates with IL-12b ($R^2 = 0.7$, $p < 0.001$). IL-10, as expected, is not expressed either at baseline or after LPS. These data have been added in supplemental figure 4.

Review Figure 1: Gene expression at baseline and after LPS treatment for select genes across the classical inbred strains of the mouse diversity panel. All strains are independently ranked by the baseline value.

Review Figure 2: Correlation of IL12b and IL6 gene expression in LPS-treated macrophages of different classical recombinant mouse strains. Linear regression. Pearson correlation, R^2 and p -value indicated.

FIG1: the authors have identified natural variation of macrophage activation in response to LPS stimulation in macrophages from humans and inbred mouse strains. However, in humans the authors used alveolar macrophages whereas in mice they used thioglycollate-elicited peritoneal macrophages. Please clarify why mouse alveolar macrophages were not used for comparison.

We thank the reviewer for bringing up this relevant point. We chose mouse peritoneal macrophages because it has been a very common source of macrophages in many studies, including studies about

macrophage polarization. The best comparison would be human peritoneal macrophages in manuscript figure 1 but such samples are difficult to obtain. Instead, we chose a small published dataset of human alveolar macrophages because of its high quality: healthy young volunteers were injected with saline or LPS in the left or right bronchus via bronchoscopy, respectively, and macrophages were after enrichment directly sequenced. Many other human monocyte-derived datasets are treated with M-CSF which likely has its own effect on inter-individual variation. We added these caveats to the discussion.

We analyzed another published human macrophage transcriptome dataset with 70 individuals (alveolar macrophages) to measure the inter-individual variation of select genes. As shown in Review Figure 3 (and new supplemental figure 3) this data also support macrophage diversity in humans. Future work should use a broader approach covering different macrophages in different organs. However, this is beyond the scope of the present study.

Review Figure 3: Variation of gene expression between human alveolar macrophages. Data includes 70 humans without LPS treatment. This data corresponds to manuscript figure 1e. The ACTB gene (beta-actin) is shown as control. Data from GSE13896.

On Page 4 in the first paragraph of the Results section in relation to FIG 1, the authors note, ‘Variation of gene expression in technical and biological replicates was low and did not explain the observed findings?’ Please include this information in the supplementary figures.

These quality controls were already published (Orozco, LD 2012) and are reproduced below (Review figure 4). Pathway activation in biological replicates is shown in Review figure 5. These data are now shown in supplemental figure 1.

Review figure 4: Biological replicates of peritoneal macrophage transcriptomes cluster mostly together (Data from Orozco, Lulis et al, 2012).

Review figure 5: NF-kB pathway activation of biological replicates. Related to figure 1. a,b) The NF-kB pathway activation was determined in macrophage transcriptomes by IPA using biological replicates of classical mouse strains.

FIG 2: Please explain why the time point of 4 hours was selected. In numerous studies, induction of Arg1 by other stimuli such as IL-4 peak between 24 and 48 hours with a delay of induction in the first 12 hours. This this early induction specific to LPS? If a time course was indeed performed, please consider including this in the Supplementary Figures.

Early induction of gene expression is typical in LPS-triggered inflammation (Nilsson R, 2006). Even after 30min to 1 hour, LPS has an early concentration-dependent effect on macrophages and monocytes activation (Meng & Lowell 1997, Arner E 2015, Guha M 2001). Therefore, for early pathway activation, many studies use only a few hours incubation when stimulated in vitro (Amura CR 1998, Arnold CE 2015). This is particularly feasible in mRNA studies, whereas cytokine measurements or functional assays can require a later time point.

Because the temporal effects of LPS are well established, we did not perform a time course experiment. We added the information in the manuscript that the 4h time point reflects an early LPS response in macrophages, and does not allow assumptions on the late response.

In FIG4 and FIG5, the authors' use of diseases correlates in interesting and a compelling correlate to the potential impact of macrophage polarization on human disease and health. Mechanisms behind strain variation in macrophage activation is fascinating yet there is no mechanistic evidence exploring these differences. Whereas this is beyond the scope of this manuscript, the authors should address various possibilities in the discussion.

We agree with the reviewer. While it is beyond the scope of this work to elucidate strain-specific mechanisms of macrophage phenotypes, it is definitively an important question to address in future studies. It is entirely possible that different mechanisms including IL6/IL10/Stat3 and different responses to LPS could lead to different or common macrophage phenotypes in these strains. We added various possibilities in the discussion.